# Genetic associations with resistance to *Meloidogyne enterolobii* in guava (*Psidium* sp.) using cross-genera SNPs and comparative genomics to *Eucalyptus* highlight evolutionary conservation across the Myrtaceae

Carlos Antonio Fernandes Santos[1]☯, Soniane Rodrigues da Costa[2]☯, Leonardo Silva Boiteux[3], Dario Grattapaglia🄳[4]*, Orzenil Bonfim Silva-Junior🄳[4]☯

1 Embrapa Semi-Arid, Petrolina, Pernambuco, Brazil, 2 Graduate program in Genetic Resources, Universidade Estadual de Feira de Santana, Feira de Santana, Bahia, Brazil, 3 Embrapa Vegetable Crops (CNPH), Brasilia, Distrito Federal, Brazil, 4 Embrapa Genetic Resources and Biotechnology (CENARGEN), Brasília, Distrito Federal, Brazil

☯ These authors contributed equally to this work.
* dario.grattapaglia@embrapa.br

## Abstract

Tropical fruit tree species constitute a yet untapped supply of outstanding diversity of taste and nutritional value, barely developed from the genetics standpoint, with scarce or no genomic resources to tackle the challenges arising in modern breeding practice. We generated a *de novo* genome assembly of the *Psidium guajava*, the super fruit "apple of the tropics", and successfully transferred 14,268 SNP probesets from *Eucalyptus* to *Psidium* at the nucleotide level, to detect genomic loci linked to resistance to the root knot nematode (RKN) *Meloidogyne enterolobii* derived from the wild relative *P. guineense*. Significantly associated loci with resistance across alternative analytical frameworks, were detected at two SNPs on chromosome 3 in a pseudo-assembly of *Psidium guajava* genome built using a syntenic path approach with the *Eucalyptus grandis* genome to determine the order and orientation of the contigs. The *P. guineense*-derived resistance response to RKN and disease onset is conceivably triggered by mineral nutrients and phytohormone homeostasis or signaling with the involvement of the miRNA pathway. Hotspots of mapped resistance quantitative trait loci and functional annotation in the same genomic region of *Eucalyptus* provide further indirect support to our results, highlighting the evolutionary conservation of genomes across genera of Myrtaceae in the adaptation to pathogens. Marker assisted introgression of the resistance loci mapped should accelerate the development of improved guava cultivars and hybrid rootstocks.

**Data Availability Statement:** The Psidium guajava genome assembly was deposited at DDBJ/ENA/GenBank under the accession JAGHRR000000000. The version described in this article is JAGHRR010000000. These resources were deposited under the BioProject ID PRJNA713343. All experimental SNP genotype data and phenotypic data for the binary and quantitative (RF) RKN resistance traits are made available in supporting S1 File.

**Funding:** This work was supported by competitive grants "NEXTFRUT" grant # 0193.001.198/2016 from Fundação de Amparo à Pesquisa do Distrito Federal (FAP-DF) to DG, CNPq (Conselho Nacional de Desenvolvimento Científico e Tecnológico) grants 485472/2012-0 e 302525/2017-3 to C.A.F. S. and Coordenação de Aperfeiçoamento de Pessoal de Nível Superior (CAPES), Finance Code 001. S.R.C. had a doctoral fellowship from Fundação de Amparo à Pesquisa do Estado da Bahia, FAPESB. C.A.F.S., L.S.B and D.G had research productivity grants from CNPq. There was no additional external funding received for this study and the funders had no role in study design, data collection and analysis, decision to publish, or preparation of the manuscript.

**Competing interests:** The authors have declared that no competing interests exist.

## Introduction

Among the economically important species of family Myrtaceae, and within the large group encompassing ~3,500 fleshy fruit species in the family [1] the guava (*Psidium guajava* L.) stands out as the most commercially relevant crop [2]. Native to tropical America and widely distributed in subtropical and tropical countries, guava is frequently referred to as a super fruit due to its high polyphenolic and multivitamin content with important ethnopharmacological properties [3, 4]. Notwithstanding its recognized nutraceutical value, the estimated worldwide annual production in 2018 was only 6.75 million tons [5], just a fraction of the 129.6 million tons of apple, for example [6]. Guava still experiences the status of a very large number of mostly unknown tropical fruits of outstanding taste and nutritional value that are either undomesticated or mostly unexploited from the genetics, genomics and breeding standpoint.

Despite the availability of extensive genetic variation in the genus *Psidium*, the genetic basis of current guava cultivars is narrow, resulting in significant susceptibility to some diseases [7]. Following the first report of its occurrence in guava in Brazil [8], the root-know nematode (RKN) *Meloidogyne enterolobii* is currently the most economically important pathogen of guava in the Neotropics [9, 10]. *Meloidogyne enterolobii* (= *M. mayaguensis*) is a highly polyphagous species with a host range similar to that of the two major RKN species, *M. incognita* and *M. javanica* [11]. However, *M. enterolobii* is a greater agricultural threat with wider virulence profile, having the ability of 'breaking down' a wide range of resistance factors effective against other major *Meloidogyne* species in many crops [11, 12].

While controlling polyphagous RKN species by crop rotation in woody perennials is not an alternative, chemical control is increasingly banned due to environmental and human health concerns. The deployment of natural plant resistance is thus the most sustainable alternative toward intensive crop production. Major advances have been made in species of *Prunus* with a suite of resistance genes to RKN species finely mapped or cloned and used in breeding for durable resistance [13, 14]. In contrast, despite the current use of resistant rootstocks, very little is known about genes and defense mechanisms underlying the RKN resistance in other mainstream woody crops such as coffee and grapevine [15]. Sources of resistance to *M. enterolobii* have been detected in wild *Psidium* relatives but not in *P. guajava* accessions [9]. The deployment of interspecific (*P. guajava* × *P. guineense*) rootstock hybrids with resistance derived from *P. guineense* Swartz is currently the only viable management strategy (Fig 1). Nevertheless, cultivar development could be more efficient by the still unexploited prospects of introgressing disease resistance genes from *P. guineense* into *P. guajava*.

Progress has been made in the Myrtaceae family by generating genomic resources for forest trees and to a much lesser extent for fruit species [16]. Major emphasis has been in *Eucalyptus* species, for which a reference genome for *E. grandis* [17] and a multi-species SNP platform EuCHIP60K [18] are available. Alike *Eucalyptus*, *Psidium* displays a chromosome number $x = 11$, a basic complement largely conserved across the family [19, 20]. Recently, a chromosome-level assembly of the *P. guajava* genome corroborated its high collinearity to the *E. grandis* genome [21]. To date several microsatellites have been published for *Psidium* [22, 23], while SNP marker data have only been generated for *Psidium guajava* using genotyping by sequencing methods based on complexity reduction with restriction enzyme digestion [24, 25]. This approach is well known to suffer from poor data reproducibility and portability across experiments, especially in heterozygous genomes [26, 27]. Conversely, cross-genera transferability of *Eucalyptus* SNPs genotyped with the gold-standard Illumina Infinium™ EuCHIP60K platform to *Psidium* had been demonstrated early on [18], opening solid possibilities of genome-wide diversity and association studies in *Psidium* using high quality and portable SNP data across genomes.

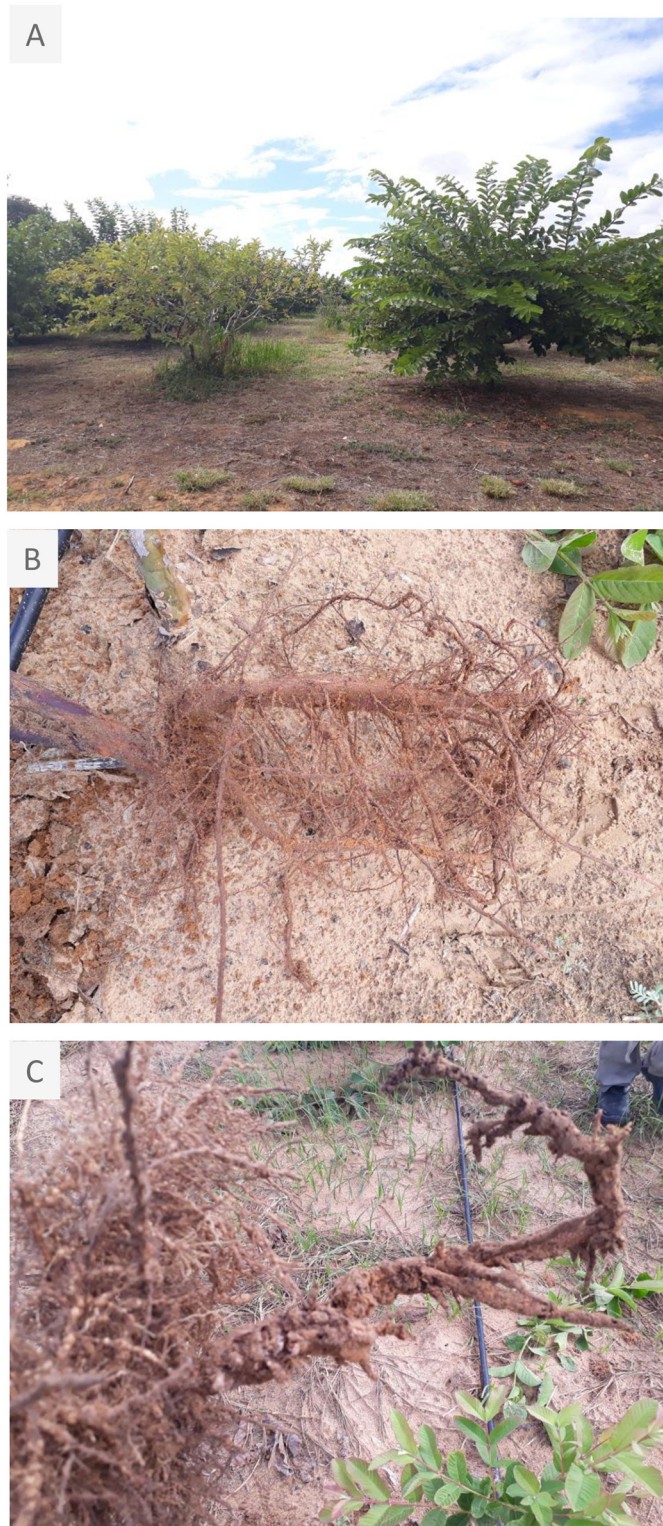

**Fig 1. (A)** *Psidium guajava* plant attacked by *M. enterolobii* root knot nematode (left) and a *P. guajava* grafted plant onto a resistant *P. guajava x P. guineense* hybrid (right); **(B)** healthy roots of resistant *P. guajava x P. guineense* hybrid and **(C)** *M. enterolobii* RKN infected roots of a susceptible *P. guajava* plant.

Association mapping has allowed important advances in mapping resistance loci to a wide array of *Meloidogyne* species across distinct hosts providing tools to assist breeding [28]. In this work, we generated a *de novo* assembly of the *P. guajava* genome and successfully transferred a large number of SNP probesets from *Eucalyptus* to *Psidium* at the nucleotide level to carry out an association study for *M. enterolobii* resistance response. Using these resources, we describe the detection of genomic regions harboring resistance loci to RKN derived from the wild relative *P. guineense*. Additionally, our results highlight the evolutionary conservation of genomes across genera of Myrtaceae in the adaptation to pathogens.

## Material and methods

### *Psidium guajava* × *P. guineense* association mapping population

To provide the best opportunity for recombination, an association mapping population was created by allowing open pollination among $F_1$ interspecific root–stock hybrids ('BRS Guaraçá' = *P. guajava* × *P. guineense*). A total of 189 outbred $F_2$ plants from open pollinated fruits harvested from 22 BRS Guaraçá $F_1$ hybrid trees were evaluated for *M. enterolobii* reaction in a controlled inoculation experiment at Embrapa Semi-Arid, Petrolina-PE, Brazil. Sample collection from native *Psidium* populations was granted by the Brazilian Institute of the Environment (IBAMA), authorization number CGEN Nº 001-B/2013.

### *Meloidogyne enterolobii* inoculation and plant reaction evaluation

Seedlings of individual $F_2$ plants (with $\approx 25$ cm in height) were inoculated with 10,000 eggs + second-stage juveniles (J2). The *M. enterolobii* inoculum was extracted from guava roots collected in a commercial area in the city of Petrolina-PE, using classic methods [29]. Each plant was inoculated with 2 mL of the suspension in each one of two hollows in the soil at a distance of 1.5 cm from the stem and 2.5 cm deep. At 120 days after inoculation, the plants were collected and the roots were carefully washed in water within a plastic container to avoid loss of nematode eggs. A qualitative binary phenotype of presence/absence of root galls and a quantitative Reproduction Factor (RF) trait were measured. Individual roots (5 grams) were processed, using a blender for 30 sec. Eggs and J2 were counted, determining the total number of nematodes (final population) and the reproduction factor (RF = final nematode population/ the initial inoculum), was calculated [30]. Plants were classified as resistant when a reduction $\geq 90\%$ in RF was observed in relation to the susceptible plants [29]. For the subsequent quantitative analysis, the RF values were transformed into $\sqrt{RF} + 0.5$ due to the frequent occurrence of individuals with zero or very low RF values.

### Genetic material and SNP genotyping

DNA extraction was performed from young leaves using an optimized protocol for high quality DNA from woody plants [31]. Plants were genotyped with the Infinium EUChip60K containing 60,904 *Eucalyptus* SNPs [18]. SNPs on the EUChip60K are coded with the chromosome number followed by the physical address in base pairs on version 1.1 of the *E. grandis* genome [17] which were then updated to the version 2.0 genome deposited at https://jgi.doe.gov/data-and-tools/. SNP genotyping was performed at GeneSeek (Lincoln, NE, USA).

### Quality control of informative SNPs

Due to the complexity of the cross-genera genotyping assay, a clustering procedure was applied to the *Psidium* derived EUChip60K intensity data. We used the standard GenCall algorithm in GenomeStudio 2.0 (Illumina, Inc. San Diego, USA) following its best practices and

criteria described earlier [18]. After these QC steps, and in order to extract the maximum amount of quality genotypic data, the optiCall algorithm was used to ascertain genotypes at common and low-frequency variants [32].

## Allelic association tests and analyses of mixed linear models

Allelic and genotypic association tests were employed, respectively, to test for the association between single SNP alleles and genotypes and the binary presence/absence of root galls and the RF phenotype. Allelic and genotypic association tests were carried out with a Fisher's exact test and Cochran-Armitage trend test, respectively. Bonferroni correction and the Benjamini & Hochberg (B&H) and Benjamini & Yekutieli (B&Y) 'false discovery rate' (FDR) procedures were implemented with PLINK [33]. A mixed linear model (MLM) analysis was carried out with the transformed RF phenotype using TASSEL 5.2.65 [34]. An 'identical by state' (IBS) matrix estimated in PLINK 1.9 [33] was used to account for the confounding effects of both population and family structure.

## Comparative genomic analyses of *Eucalyptus* associated SNPs in the *Psidium* genome

At the time of this study, a genome sequence assembly was not available for *P. guajava* or *P. guineense*. To allow bona fide comparative genomic analyses of the associated genomic regions using *Eucalyptus* SNPs onto the *Psidium* genome, we sequenced the genome of the *Psidium guajava* $S_2$ plant UENFGO8.1–10 derived from two generations of selfing using PACBIO Sequel I technology. The FALCON-Unzip v.1.1.5 pipeline [35] was applied to produce an initial assembly. Contigs were aligned to the *Eucalyptus grandis* v2.0 genome sequence in Phytozome v13 [36] using the Cactus whole-genome multiple alignment program [37]. To obtain the full value of the Cactus protocol and to mitigate the effect of the fragmentary assembly on alignment quality, we aligned the contigs to a set of three chromosome-level related genomes. We included the *E. grandis* (Myrtaceae) and the most likely sister to Myrtaceae, the *Punica granatum* (Lythraceae), as well as *Vitis vinifera* (Vitaceae) as the ultimate outgroup. The tree topology was generated starting from proteome data to reconstruct phylogenies that chart the relationships among these organisms. Protein sequences into the genomes were downloaded from the NCBI repository. For *P. guajava* proteins, the Augustus pipeline [38] was run on the contig assembly using the extrinsic evidence provided by aligning RNA-Seq data available for the species in NCBI SRA. Single-copy genes in the BUSCOv3 program [39, 40] were used to identify shared subsets from the different sets of protein data across the genomes. Proteins were aligned using MAFFT and filtered with trimAl, and the maximum likelihood tree was built using RAxML [39, 41] to estimate the species phylogeny. The final non-overlapping whole-genome multiple alignments with ProgressiveCactus described as a HAL file [42] were extracted for the eleven chromosomes of the *E. grandis* and linked further using the algorithm implemented in Ragout2 [43].

The reference genome assembly using synteny in Ragout2 was used to ensure consistency of the transference of the SNP probesets from the target genome of the *P. guajava*. For this purpose, starting from a BED file with probesets coordinate annotation on the *E. grandis* genome, we used the halLiftOver procedure in the HAL package to perform a base-by-base mapping between guava and *Eucalyptus*. The output which refers to the probeset sequences in the *E. grandis* genome to their corresponding locations in guava was written in the PSL format and then converted to the AXT format using the program utilities in the Kent's Utilities package from the UCSC Genome Browser tools [44]. The corresponding locations and sequence content of probesets in guava were inspected to accept only those that matched the same

original positions in the query genome and recovered the same assayed substitution at the nucleotide-level. Otherwise, the probeset location was called erroneous and discarded. Finally, we performed a transcript consistency analysis using the comprehensive transcript set provided for the *E. grandis* genome assembly. We used the CAT pipeline [45] for each of one of 46,280 transcripts in the *E. grandis* annotation to determine their contig location and orientation in the contigs into the *P. guajava* genome with respect to the alignments between the two genomes from the HAL file and extrinsic evidences based on public mRNA-Seq data. After the comparative annotation of protein-coding genes, an evaluation of the putative effects of genetic variants was carried out using a database prepared using the SnpEff v5.0c pipeline [46].

## Results

### EUChip60K performance in *Psidium*

A total of 6,879 SNPs were successfully assayed from the 60,904 EuCHIP60K SNPs using the GenCall clustering algorithm with a call rate CR$\geq$ 0.90 and no minimum allele frequency (MAF) threshold. The normalized X-Y intensity data for these 6,879 SNPs were exported from Genome Studio (GS) and genotypes ascertained using the optiCall algorithm (default options) resulting in 6,225 SNPs. When a MAF $\geq$0.01 was applied to the 6,879 SNPs only 521 SNPs were retained with the standard GenCall algorithm in GS, while optiCall delivered 4,143 polymorphic SNPs, an eight-fold improvement (S1 File).

### Phenotypic evaluation

Out of the 189 originally inoculated plants, 175 ultimately survived to final evaluation. For the binary phenotype, 92 resistant plants were free of *M. enterolobii* induced galls while 83 plants displayed conspicuous root galls and were classified as susceptible. The untransformed Reproduction Factor (RF) phenotype varied from zero to 2.53 and when considered in a simple binary fashion only six of the 175 plants were scored with a reproduction factor RF$>$ 1.0, thus classified as susceptible, while all others, with RF $<$1.0, were rated as resistant (S1 File). Although these binary counts fitted the same epistatic models proposed earlier in an inbred $F_2$ population [47] only one out of the 4,143 SNP data segregated in a 1:2:1 ratio indicating that the population did not behave as a regular inbred $F_2$ from a single $F_1$ plant but rather, as expected, as a mixture of outbred crossed offspring.

### Allelic and genotypic association analyses

Following multiple test corrections three and four SNPs displayed significant allelic and genotypic association respectively for the binary trait of *M. enterolobii* galls according to the Cochran-Armitage trend test (with adjusted p-value $<$ 5.0E-3) (Table 1). The combination of allelic and genotypic association tests indicated two SNPs (*viz.* EuBR03s29615246 and EuBR03s30383415) in significant association with the overall resistance reaction. The SNP EuBR03s30383415 found significant by both the allelic and genotypic association, was later found significantly associated by the MLM analysis as well. The additional SNPs also on chromosome 3, EuBR03s16993500, EuBR03s37875650, EuBR03s21599380 showed variable signal depending on the statistical association test employed and were not considered further.

### Association mapping with a mixed linear model

The MLM analysis incorporating population and kinship covariates resulted in one SNP (EuBR03s30383415) associated with *M. enterolobii* resistance using TASSEL following the specified threshold p-value of $<$5.0E$^{-04}$, with coefficient of determination ($R^2$) of 0.107

**Table 1. SNP markers displaying allelic and genotypic associations and corresponding adjusted p-values for the binary trait presence/absence of *Meloidogyne enterolobii* galls in 175 outbred F$_2$ plants derived from open pollination among F$_1$ hybrid *Psidium guajava* × *P. guineense* plants.** SNPs are coded with the chromosome number followed by the physical address in base pairs on version 2.0 of the *E. grandis* genome.

| SNP marker | Chromosome | Adjusted p-value | | |
|---|---|---|---|---|
| | | **Bonferroni** | **FDR/B&H**[*] | **FDR/B&Y**[**] |
| **Allelic association—Fisher's exact test** | | | | |
| EuBR03s29615246 | 03 | 6.26E$^{-16}$ | 6.26E$^{-16}$ | 5.21E$^{-15}$ |
| EuBR03s16993500 | 03 | 1.48E$^{-05}$ | 7.40E$^{-06}$ | 6.16E$^{-05}$ |
| EuBR03s30383415 | 03 | 0.002022 | 0.000674 | 0.005616 |
| EuBR03s37875650 | 03 | 0.05376 | 0.01344 | 0.112 |
| EuBR03s11126682 | 03 | 0.5225 | 0.1045 | 0.8705 |
| **Genotypic association—Cochran-Armitage trend test** | | | | |
| EuBR03s29615246 | 03 | 2.84E$^{-07}$ | 2.84E$^{-07}$ | 2.37E$^{-06}$ |
| EuBR03s37875650 | 03 | 0.000289 | 0.000145 | 0.001203 |
| EuBR03s30383415 | 03 | 0.001052 | 0.000351 | 0.002921 |
| EuBR03s21599380 | 03 | 0.001934 | 0.000484 | 0.004028 |
| EuBR03s16993500 | 03 | 0.03814 | 0.007628 | 0.06355 |
| EuBR03s7520846 | 03 | 0.05023 | 0.008372 | 0.06974 |

[*]False discovery rate (FDR) Benjamini & Hochberg (B&H).

[**]False discovery rate (FDR) Benjamini & Yekutieli (B&Y).

(Fig 2; Table 2), suggesting that factors residing in this genomic region may explain a considerable fraction of the *M. enterolobii* resistance phenotype. Three additional SNPs were suggestive of association, one also on chromosome 3 and the other two on different chromosomes. Two of the four SNPs detected in the MLM analysis, namely EuBR03s30383415 and EuBR03s16993500, both on chromosome 3, had also been detected by both the allelic and the genotypic association analyses.

## Genome assembly of *P. guajava* and comparative genomic analyses to the *E. grandis* genome

The assembled contigs for our whole genome shotgun of *P. guajava* totaled 359.4 Mb into 510 sequences and covered 78% of the 463 Mb genome size estimated for *P. guajava* by flow cytometry [19]. This assembly was deposited at DDBJ/ENA/GenBank under the accession JAGHRR000000000. The version described in this article is JAGHRR010000000. These resources were registered under the BioProject ID PRJNA713343. These contigs were further linked to provide a reference genome assembly from the whole-genome alignments against the *E. grandis* assembly using a syntenic path approach. Some of the original contigs were broken to fit into the hierarchical block construction, suggesting the presence of rearrangements of the homologous sequences between the two genomes across large segments (>10 kbp). At smaller distances in the range of the probeset sequence length (100 bp) the resulting syntenic path covered 32.1% of our *de novo* guava's assembly, which allowed us to successfully reallocate 14,268 probesets of the EucHIP60k. These probesets are distributed across 377 contigs (328 Mb) into the assembly. Comparative transcript identification resulted in the annotation of 21,240 protein-coding loci. Importantly, out of the 4,143 SNP probesets that detected polymorphism, 1,241 were fully converted between the genomes at the nucleotide-level resolution and matched the original allelic variation of the SNP into the genotyping assay. A detailed analysis was therefore possible for these variant positions in the *P. guajava* genome. Prediction of effects at these genomic variants revealed diverse types of impact on 6,201 transcripts in the

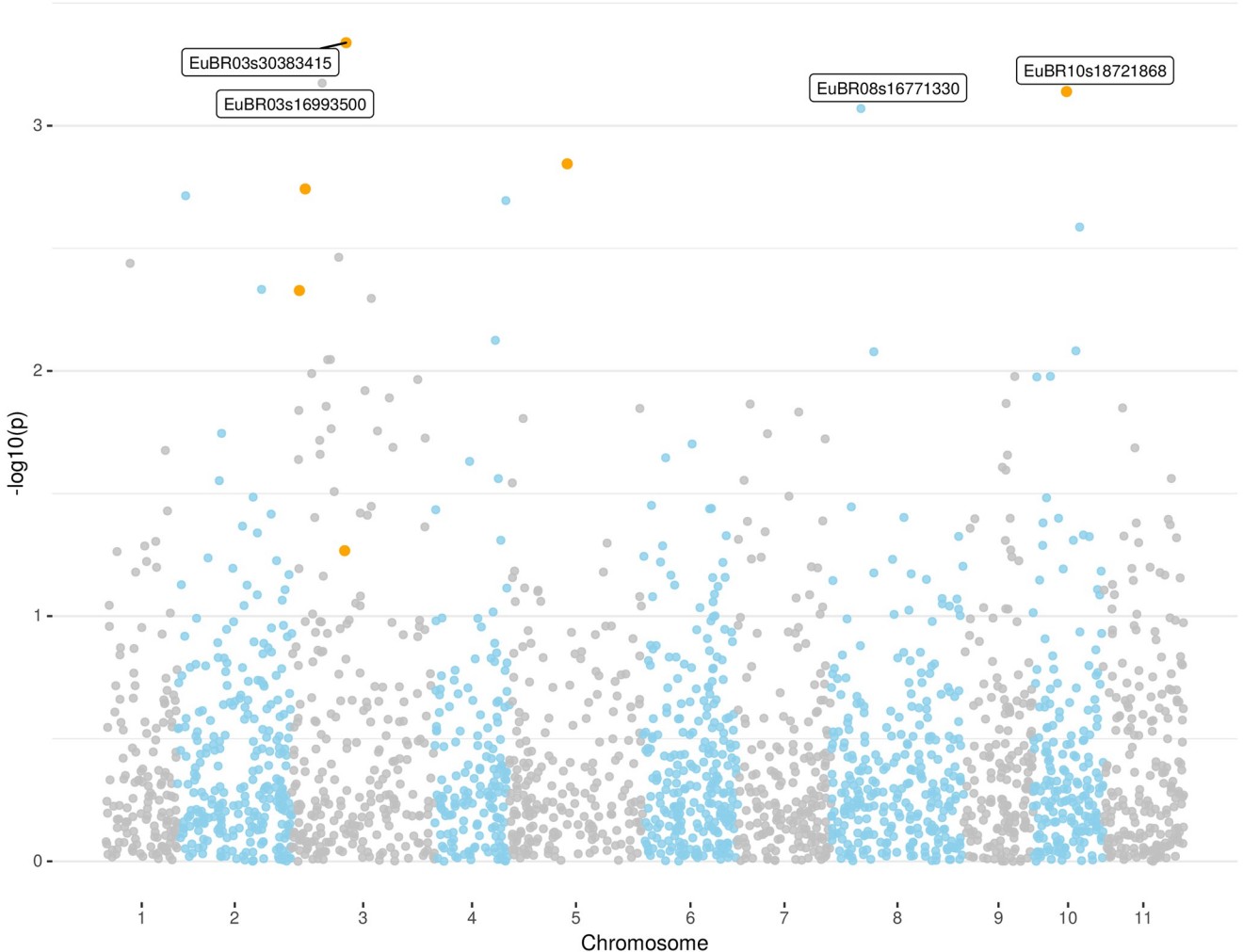

**Fig 2. Manhattan plot of the association analysis for *M. enterolobii* root knot nematode (RKN) resistance in guava from a MLM analysis.** Significantly associated SNPs (see Table 2) are labeled and those for which SNP probesets were reallocated on the *Psidium guajava* genome are highlighted by orange dots.

*P. guajava* genome from which 2,379 have putative homologues in the *E. grandis* genome assembly. A SNP annotation file (VCF) (S2 File) and an Illumina GenomeStudio™ 2.0 cluster file (EGT) for the successfully reallocated probesets are provided (S3 File). The EGT file will be particularly valuable for processing and quality control of SNP data for future *Psidium* genotyping experiments using the EucHIP60K.

**Table 2. SNPs displaying significant associations and their coefficients of determination ($R^2$) according to a mixed linear model (MLM) analyses in TASSEL for reproduction factor of *Meloidogyne enterolobii* root know nematode in 175 outbred $F_2$ plants derived from open pollination among $F_1$ hybrid *Psidium guajava* × *P. guineense* plants.** SNPs are coded with the chromosome number followed by the physical address in base pairs on version 2.0 of the *E. grandis* genome.

| SNP | *Eucalyptus/Psidium* chromosome | p-value | $R^2$ |
|---|---|---|---|
| EuBR03s30383415 | 03 | 4.59E$^{-04}$ | 0.107 |
| EuBR03s16993500 | 03 | 6.70E$^{-04}$ | 0.097 |
| EuBR10s18721868 | 10 | 7.26E$^{-04}$ | 0.094 |
| EuBR08s16771330 | 08 | 8.51E$^{-04}$ | 0.115 |

The probe sequences for the *M. enterolobii* resistance associated SNPs EuBR03s29615246 and EuBR03s30383415 spaced by 768 kbp on the *Eucalyptus* genome were found spaced nearly 470 kbp apart onto a single sequence of the *P. guajava* assembly that spans 3.14 Mbp in size (JAGHRR010000008.1). The physical interval covered by this contig in the pseudo-assembly of the *Psidium guajava* using the syntenic path alignment approach to the *Eucalyptus grandis* assembly corresponds to a genomic region of about 7.45 Mb along Chromosome 3 (coordinates 27,580,466 to 35,980,336 of the Egrandisv2.0).

Our pipeline for protein-coding gene annotation predicted 102 gene models with coordinates within the contig of the *Psidium guajava* assembly containing the probes for SNPs EuBR03s29615246 and EuBR03s30383415. The gene density of 32.5 genes/Mbp is similar to its corresponding putative orthologous region in the *Eucalyptus grandis* assembly, which has a density of 36.1 genes/Mbp (269 gene models distributed along 7.5 Mbp). Interestingly, the comparison of the repertoire of putative NBS-LRR genes between the genomes along this orthologous region shows a different picture. While the *Eucalyptus* genome contains 31 models of physical arrangement in this gene family within this locus, we could identify only 2 models within the corresponding region in the *Psidium guajava* assembly.

In terms of the predicted impact based exclusively on a bioinformatics analysis, these two SNPs are variants with moderate effect causing non-synonymous changes in respect to the conceptual translation of the transcripts J3R85_001472 /Eucgr.C01744 (Uniprot id: A0A059CPK6) and J3R85_001443/Eucgr.C01791 (Uniprot id: A0A059CPT9), respectively. SNP EuBR03s30383415 is also a modifier variant occurring downstream to the gene locus for J3R85_001444/ Eucgr.C01790 (Uniprot id: A0A059CQ27). The physical location, corresponding probesets and features of the predicted variant genes for these two associated SNPs on chromosome 3 are summarized (Fig 3; Table 3) and further discussed below.

## Discussion

### The EuCHIP60K provides genome-wide SNP genotyping in *Psidium*

Our genetic association experiment was driven by successful genotyping and precise reallocation of SNP probesets from the *Eucalyptus* genome to a *de novo* assembly of the *Psidium* genome. Typically, for poorly funded orphan crops, SNP genotyping has been carried out by one of the several methods of restriction enzyme-based reduced representation sequencing [48]. This approach has recently been used to investigate the genetic diversity of *Psidium guajava* [24], as well as related species of the genus [25]. These methods offer the advantage of simultaneous SNP discovery and genotyping with no upfront costs, but suffer from well-known pitfalls and limitations in data quality, reproducibility and especially portability across experiments, particularly for highly heterozygous genomes [26, 48, 49]. Although SNPs datasets are generated, the ascertained SNPs vary across experiments and as such they do not constitute a true legacy genomic resource for future widespread and long-term use by the community, nor they allow consolidation of data across studies in the same manner as the fixed content chip-based SNPs data and resource we have described and provided in this work.

Although fixed content chips are currently the gold standard for SNP genotyping, successful attempts to transfer platforms across species and genera have been limited to domestic animals [50], showing a linear decrease of 1.5% in SNP call rate per million years divergence and exponential decay of polymorphisms retention [51], consistent with theoretical expectations [52]. In plants, to the best of our knowledge no large-scale transferability evaluations of SNP arrays across genera have been reported. The 6,879 out of 60,904 SNPs successfully transferred from *Eucalyptus* to *Psidium* in this work corroborate our earlier estimates of ~10%

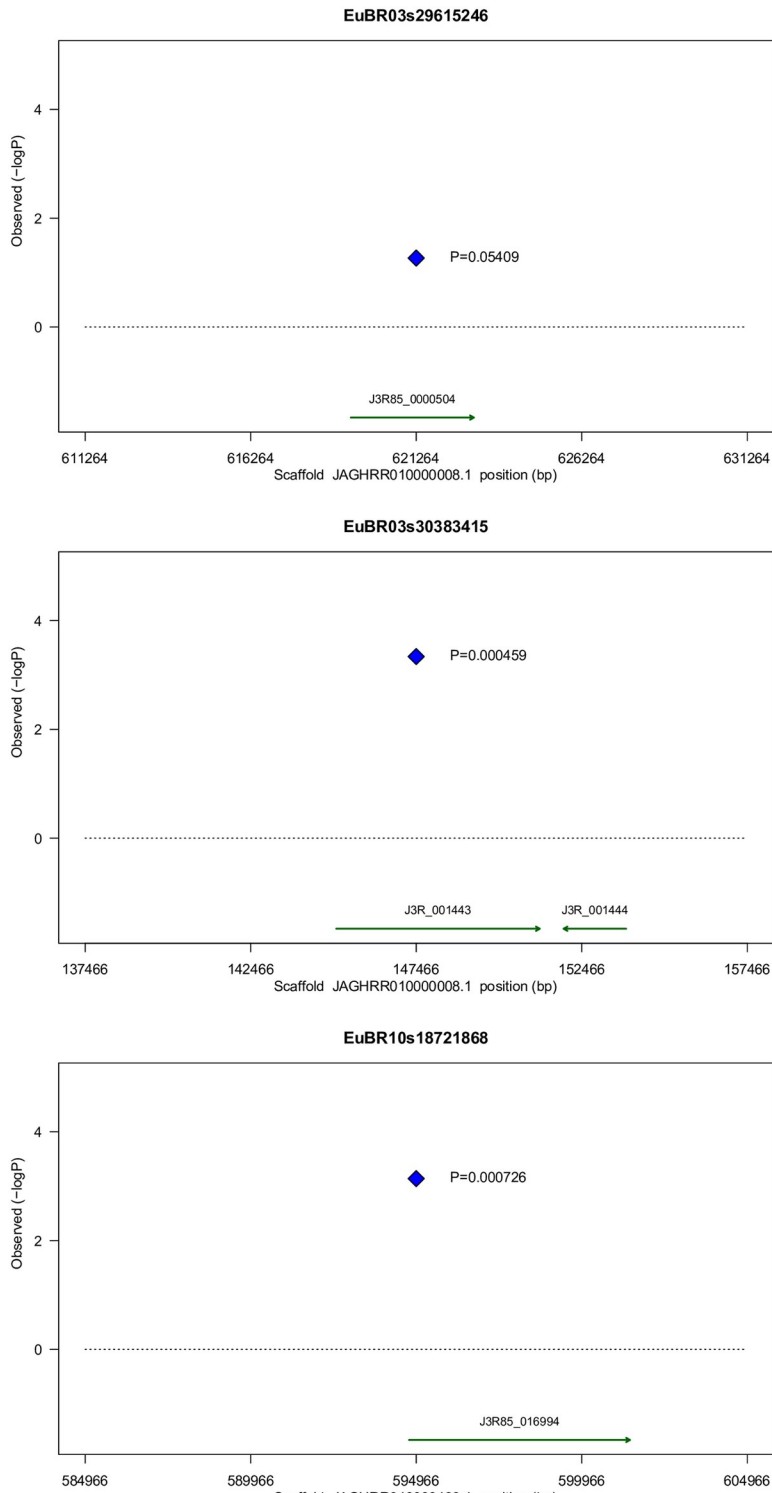

**Fig 3. Schematic diagrams of the physical scaffold location of the three main SNPs on chromosomes 3 and 10 of the *Psidium guajava* genome associated with RKN resistance with their corresponding p-values and associated functionally relevant defense genes annotated (see text and Table 3 for details).** Coordinates are centered at the SNP (blue diamond) and expand 10 Kbp to both sides.

**Table 3. Summary of the main features of two SNP variants associated with RKN resistance on chromosome 3 and chromosome 10 of *Psidium guajava* likely to cause functional or regulation changes on predicted genes (see text for details).**

| Feature | EuBR03s3083415 | EuBR03s29615246 | EuBR10s18721868 |
|---|---|---|---|
| *P. guajava* assembly scaffold | JAGHRR010000008.1 | JAGHRR010000008.1 | JAGHRR010000188.1 |
| *P. guajava* assembly locus | 147,466 | 621,264 | 594,966 |
| Reference allele | G | G | C |
| Major allele (A) associated with the RKN resistant phenotype) | A | A | T |
| Minor allele (B) associated with the RKN susceptible phenotype | G | G | C |
| Minor allele frequency | 0.253 | 0.467 | 0.271 |
| Counts of genotype AA | 102 | 86 | 97 |
| Counts of genotype AB | 57 | 9 | 56 |
| Counts of genotype BB | 6 | 71 | 8 |
| Downstream gene variant | ncbi: J3R85_001444; uniprot:EUGRSUZ_C01790 | - | - |
| Intron variant | - | - | - |
| Missense variant | ncbi: J3R85_001443; uniprot ortholog:EUGRSUZ_C01791 | ncbi: J3R85_001472; uniprot ortholog:EUGRSUZ_C01744 | ncbi:J3R85_0015914; uniprot ortholog:EUGRSUZ_J01461 |
| synonymous_variant | | | |
| Protein name | EUGRSUZ_C01790: ENT domain-containing protein; EUGRSUZ_C01791:DUF642 | EUGRSUZ_C01744: F-box domain-containing protein | EUGRSUZ_J01461: Histone acetyltransférase |
| SNP probe sequence [variant SNP site] | GTGCCCTATGAGTCGRAGGGCAAAGGCGGGTTCAAGCGCGCTGTCCTGCGGTTCCAGGCC [A/G] TGTCCATGAGGACCAGGATCATGTTCTACAGCACGTTCTACACCATGAGGAGTGACGATT | ATTGGGCACTATGCCACGGCTGTTGCTGATTTAACCCTTACTAGCCTCCATAATGTCACT [A/G] AGAGAGGGCTTTGGGTCATGGCAATGGTCATGGTTTGCAAAGGTTGAGGTCTTTGATAG | CGTACAATGTACATTCTGCCCATCGTGCCACAATTGTTGCAGAGGTGGGACCATCAAGTG [C/T] GGGGGGATTTCTGTTTCCAAGGCAAGCTCCACAAGGAAGAGAAGAATAATGAGGAAATTGAA |

transferability [18] consistent with the estimated divergence time of 60–65 Million years (90% decrease in SNP calling rate) between tribes Eucalypteae and Myrteae [53], with a slightly higher than theoretically expected polymorphism retention of 6.8% (4,143 SNPs in 60,904) following data clustering with the optiCall algorithm. Although almost 7,000 SNPs, out of which 4,143 were informative in this particular population, might seem a small number, it represents a substantial increase in high quality marker availability for *P. guajava*, opening great prospects for genetic and breeding applications. We speculate that this higher-than-expected SNP retention might be due to the deliberate design of the Eucalyptus multispecies EUChip60K toward conserved genomic regions [18], likely subject to stronger purifying selection allowing better opportunity for polymorphism retention of targeted sites.

### *M. enterolobii* resistance in *P. guajava* reveals evolutionary conservation of chromosome 3 loci across genera of Myrtaceae

Despite the limited power of our genetic association experiment, all analytical approaches consistently pointed to a clear-cut signal involving two SNPs (EuBR03s29615246 and EuBR03s30383415) associated with *M. enterolobii* resistance on a specific sequence of the assembly of *Psidium guajava* (JAGHRR010000008.1) syntenic to chromosome 3 of *Eucalyptus grandis* (Chr03:27,580,466–35,980,336). This result is supported by the accurate sequence-level reallocation of SNP probesets from *Eucalyptus* chromosome 3 on our *de novo P. guajava* genome assembly, and corroborated by the highly conserved syntenic relationship between these two genomes [21]. In fact, a number of studies in *Eucalyptus* have reported disease resistance loci for different fungal pathogens also pathogenic in *Psidium* on chromosome 3 [54–59]. Genome annotations have also highlighted the highest densities of clusters and superclusters of NBS-LRR (nucleotide binding site-leucine-rich repeat) resistance genes on *Eucalyptus* chromosomes 3, 5, 6, 8 and 10 [60]. Observations at the gene level have shown extensive syntenic blocks of 778 genes in 522 gene families shared between *Eucalyptus* chromosome 3 and *Populus* chromosome XVIII, with the most common family represented by 33 disease resistance genes [17]. All these evidences not only provide further indirect support to our results, but also contribute to highlight a seemingly strong evolutionary conservation of the role of chromosome 3 across genera of Myrtaceae and possibly beyond, in the adaptation to different pathogens. Furthermore, the detection of major effect loci encompassing regions with NBS-LRR genes has been a common feature for resistance to RKNs across many pathosystems of both annual [28] and woody perennial [15] crops. To the best of our knowledge, our report is the first for a species in the large family Myrtaceae.

### Associated SNPs are located within or in close proximity to functionally relevant defense genes in the *Psidium* genome

The precise nucleotide-level reallocation of *Eucalyptus* SNP probesets onto the *Psidium* genome and functional effect prediction, allowed an in-depth examination of the two SNPs located in gene loci likely linked to the defense response on chromosome 3. This, in turn, allowed proposing plausible molecular mechanisms underlying the resistance response at the gene level with an emphasis for the SNPs on chromosome 3 (Fig 3; Table 3). Although we are fully aware that conclusive proof will require additional experimental evidence beyond the scope of this initial SNP discovery, we contend that the following discussion provides a potential roadmap to guide follow-up experiments. Furthermore, we tested and excluded the hypothesis that the gene loci underlying the significant SNPs could be orthologs to the *Ma* gene, a large-spectrum *Meloidogyne* species resistance locus described in *Prunus* linkage group

7 (Prupe.7G065400) [14]. Thus, the gene loci discussed below more likely represent novel sources of RKN resistance, possibly specific to Myrtaceae.

SNP EuBR03s29615246 is located in the gene J3R85_001472 and its *Eucalyptus* ortholog gene Eucgr.C01744 (Uniprot id: A0A059CPK6). This gene codes for a F-box protein belonging to the leucine-rich repeat (LRR) family involved in cell cycle control and glucose signaling, with orthology relationship to Arabidopsis EBF1/EBF2 [61]. In Arabidopsis, EBF1 and EBF2 play a specific role in the recognition of the EIN3 (ethylene-insensitive3) transcription factor (s) and facilitate their subsequent SCF-dependent ubiquitylation and degradation via the ubiquitin/26S proteasome pathway [62]. This pathway is established as an important posttranscriptional mechanism that allows eukaryotic cells to respond rapidly to signal molecules and changes in environmental conditions [63].

On the same contig where EuBR03s29615246 resides, the second SNP, EuBR03s30383415, was detected as associated. This SNP was declared significant by all analytical approaches, ranked as the strongest association with the lowest p-value in the MLM analysis (Tables 1 and 2; Fig 3). This SNP results in a non-synonymous change on the product of the gene J3R85_001443, which is an ortholog of Eucgr.C01791 (Uniprot id: A0A059CPT9) in *Eucalyptus*. Its product is a protein that contains a DUF642 domain and a galactose-binding-like domain fold, and it is member of a group of seven proteins in *Eucalyptus* that have orthology relationships to a group of ten proteins in *Arabidopsis*. Remarkably, two members of this group are AT4G32460 (BIIDXI locus) and AT2G41800 (TEEBE locus), recently reported as highly induced genes by auxin during early interaction between the susceptible *A. thaliana* ecotype Columbia and the RKN *M. incognita* [64]. SNP EuBR03s30383415 is also predicted to cause an allelic change in the region downstream of another gene of *P. guajava*, J3R85_001444, which has orthology relationship to Eucgr.C01790 (Uniprot id: A0A059CQ27) in *Eucalyptus*. This gene codes for a protein containing an EMSY N-terminal domain (PF03735), a central Agenet domain (PTHR31917), and a probable coiled-coil motif at the C-terminus. The Agenet domain is member of the superfamily Agenet/Tudor [65] and in plants it was suggested to act as a link between DNA replication, transcription and chromatin remodeling during flower development [66], a process involving hormonal regulation in which auxin plays a major role [67].

Based on these findings, we speculate about the possibility of SNPs EuBR03s29615246 and EuBR03s30383415 being important leads to identify underlying genes in the *Psidium guajava* assembly, highly syntenic to chromosome 3 in *Eucalyptus*, whose expression is likely to undergo changes during pathogen invasion due to effects of chemical modulation of plant hormone levels or signaling contributing to the resistant response in P. *guajava* × P. *guineense* plants. Although this hypothesis awaits gene expression experiments to be tested, we have advanced our discussion based on available evidences regarding the disease status of RKN infected trees.

### *M. enterolobii* resistance might be triggered by mineral nutrients and phytohormone homeostasis or signaling subject to miRNAs modulation

Studies in guava [68], coffee [69], peach and almond [70] have shown that disease status of RKN infected trees is related to symptoms triggered by nutritional imbalances in the concentration of nitrogen, calcium, manganese and magnesium in several tissues in adult plants. Furthermore, because our binary phenotypic data fitted the same epistatic models proposed earlier [47], we further considered the hypothetical involvement of a second genetic locus interacting with the underlying genes on chromosome 3. Particularly, the associated SNP EuBR10s18721868 (Table 2; Fig 3) residing on the 700 kb long sequence JAGHRR010000188.1

of the *P. guajava* assembly looked promising enough in demanding further investigation. EuBR10s18721868 is a synonymous variant inside of gene J3R85_016994 whose ortholog in *Eucalyptus* is Eucgr.J01461 on chromosome 10. This gene encodes a CREB-binding protein (CBP)/p300 protein of the subfamily of highly conserved histone acetyltransferase (HAT) and histone deacetylase (HDAC). These proteins are involved in various physiological events and their homologs in plants, the HAC genes, were recently suggested to be involved in ethylene signaling [71].

Studies in animals, plants, and viruses have suggested that microRNA function may affect synonymous codon choices in the vicinity of its target sites [72]. Following that suggestion, we performed a preliminary search for individual miRNA in the scaffold JAGHRR010000188.1, that resulted in two different families of miRNA namely miR172 and miR393. MicroRNA miR393 has been shown to take part in plant metal homeostasis, uptake and accumulation of various nutrient ions under low-nutrient conditions including nitrogen and divalent cations [73, 74]. microRNA identification on this sequence showed the highest number of loci for miR172 and it also contains one of the only two loci that were found to code for miR393 in the genome assembly of *P. guajava*.

Interestingly, nitrogen supply has shown to promote the upregulation of miR393 for targeting auxin receptor genes that encode F-box proteins such as that encoded by the gene J3R85_001472 impacted by the SNP EuBR03s29615246 on chromosome 3. These proteins are involved in ubiquitin-mediated degradation of specific substrates during auxin signaling cations [74]. Moreover, miR393 targets transcripts that code for basic helix-loop-helix (bHLH) transcription factors and for the auxin receptors TIR1, AFB1, AFB2, and AFB3. Studies have shown that miR393/AFB3 is a unique N-responsive module that controls root system architecture in response to external and internal N availability in Arabidopsis [75]. The miR172 family has a recognized role in damping the expression of genes encoding the APETALA2 (AP2) transcription factor members of a large family of the AP2/EREBP family from *Arabidopsis thaliana* and other plants, which are responsible in part for mediating the response to ethylene [74].

Although we are aware that the preliminary analysis described above lacks direct supportive experimental data to ascertain the impact of miRNAs on *Psidium* response to RKN infection, these conjectures fit both the epistatic interaction model described previously [47] and the emerging view that specific developmental or stress events can be frequently subject to modulation by diverse miRNA families [74]. Therefore, it could be postulated that this region on chromosome 10 might constitute a distinct locus epistatic to the genetic locus on chromosome 3 harboring SNPs EuBR03s29615246 and EuBR03s30383415 potentially contributing to the *M. enterolobii* resistance response.

## Concluding remarks

We mapped two genomic regions associated with resistance to *M. enterolobii* in a member of the Myrtaceae. Comparative genomics driven by significantly associated SNPs and a *de novo* assembly of the *P. guajava* genome support our conclusion that the continuous physical stretch of nearly 3.14 Mbp, encompassing a single sequence of *Psidium guajava* largely syntenic to *Eucalyptus grandis* chromosome 3, harboring SNPs EuBR03s29615246 and EuBR03s30383415, may constitute a resistance locus for RKN response derived from the wild relative *P. guineensis*. Detailed functional annotations and positioning of the gene loci targeted by these SNPs allowed us to provide a roadmap to putative underlying molecular mechanisms of resistance response at the gene level, to guide follow-up experimental work. Further indirect support to our data comes from a number of previous reports mapping resistance loci in the

syntenic genomic region in *Eucalyptus*, a region housing the highest density of NBS-LRR resistance genes and showing elevated synteny and gene retention in the genome evolution of woody plants. Additional SNPs were detected on other chromosomes suggesting that other loci besides those on chromosome 3 likely participate in the response, notably on chromosome 10, also supported by indirect evidences from *Eucalyptus* mapping studies.

Controlling RKN species by the introgression of resistance loci represents a promising alternative to environmentally undesirable nematicides. This has been the preferred approach in a growing number of mainstream woody crops, such as *Prunus*, grapevine and coffee. Our work represents the first association study in a fruit crop of the large group of fleshy members of the Myrtaceae family, engaging the guava fruit in this select group of fruit crop that have been the subject of genomic investigation. Our contribution should advance the development of new guava cultivars through the development of inexpensive SNP based assays to monitor the introgression of the *P. guineense*-derived resistance into guava cultivars and rootstocks, and foster comparative genomic studies in other species of the genus that endure the damaging effects of this pathogenic nematode.

## Supporting information

**S1 File. Phenotypic data for the binary and quantitative (RF) RKN resistance traits and SNP genotype data for the 4,143 probesets in the EucHIP60K that have passed quality control using Genome Studio and displayed polymorphism following the optiCall analysis (SNP Call Rate $\geq$90% and MAF $\geq$0.05).** Normalized X-Y data for the 6,879 successfully assayed SNPs (Call Rate $\geq$90%) were exported from GS and genotypes were ascertained using the optiCall algorithm (default options).
(XLSX)

**S2 File. SNP annotation file (VCF format) for the 14,268 *Eucalyptus* probesets in the EucHIP60K successful reallocated onto the *Psidium* guajava genome assembly using whole-genome alignment and extraction of synteny blocks.** This file highlights the 1,241 SNPs that were polymorphic in our study (GT ="0/1") and also displays the functional effect predictions resulting from the allelic changes at these genomic variants on the protein-coding loci in the *P. guajava* assembly using SnpEff terms (ANN field). Homology relationship with the protein-coding loci in the *E. grandis* assembly was inferred with the Comparative Annotation Toolkit program.
(VCF)

**S3 File. Diploid Cluster file (EGT format of Illumina GenomeStudio™ 2.0 software) for the EucHIP60K Infinium assay for *Psidium sp*. genotyping.** Cluster positions were derived from a set of 70 diverse samples of *Psidium guajava*, *Psidium guineensis* and their $F_1$ hybrids using the GenTrain algorithm. It corresponds to the SNP data from the 14,268 successful reallocated probesets onto the *P. guajava* assembly while the remaining SNP data were "zeroed". This file is provided to help processing and quality control of SNP data for future *Psidium sp*. genotyping using the EucHIP60K.
(EGT)

## Acknowledgments

We would like to thank Dr. Alexandre Pio Viana, from the State University of the North Fluminense Darcy Ribeiro for providing leaf tissue of the *P. guajava* selfed $S_2$ plant UENFGO8.1–10 used to generate the *de novo P. guajava* genome assembly. We also wish to thank Dr.

Priscila Grynberg and Dr. Marcos Costa, from Embrapa Genetic Resources and Biotechnology, for carrying the preliminary bioinformatics analysis on the miRNA identification.

## Author Contributions

**Conceptualization:** Carlos Antonio Fernandes Santos, Leonardo Silva Boiteux.

**Data curation:** Soniane Rodrigues da Costa, Orzenil Bonfim Silva-Junior.

**Formal analysis:** Carlos Antonio Fernandes Santos, Orzenil Bonfim Silva-Junior.

**Funding acquisition:** Carlos Antonio Fernandes Santos, Dario Grattapaglia.

**Investigation:** Carlos Antonio Fernandes Santos, Soniane Rodrigues da Costa, Leonardo Silva Boiteux, Dario Grattapaglia.

**Methodology:** Carlos Antonio Fernandes Santos, Soniane Rodrigues da Costa, Orzenil Bonfim Silva-Junior.

**Project administration:** Dario Grattapaglia.

**Resources:** Carlos Antonio Fernandes Santos, Leonardo Silva Boiteux, Dario Grattapaglia.

**Software:** Orzenil Bonfim Silva-Junior.

**Validation:** Leonardo Silva Boiteux, Orzenil Bonfim Silva-Junior.

**Writing – original draft:** Carlos Antonio Fernandes Santos, Leonardo Silva Boiteux, Dario Grattapaglia, Orzenil Bonfim Silva-Junior.

**Writing – review & editing:** Dario Grattapaglia, Orzenil Bonfim Silva-Junior.

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
