## [Decision Letter · Decision Letter 0]

6 Oct 2022

PONE-D-22-23102Genetic associations with resistance to Meloidogyne enterolobii in guava (Psidium sp.) using cross-genera SNPs and comparative genomics to Eucalyptus highlight evolutionary conservation across the MyrtaceaePLOS ONE

Dear Dr. Grattapaglia,

Thank you for submitting your manuscript to PLOS ONE. After careful consideration, we feel that it has merit but does not fully meet PLOS ONE’s publication criteria as it currently stands. Therefore, we invite you to submit a revised version of the manuscript that addresses the points raised during the review process.

We look forward to receiving your revised manuscript.

Kind regards,

Zhenhai Han, PhD

Academic Editor

PLOS ONE

Journal Requirements:

"This work was supported by competitive grants "NEXTFRUT" grant # 0193.001.198/2016 from Fundação de Amparo à Pesquisa do Distrito Federal (FAP-DF) to DG, CNPq (Conselho Nacional de Desenvolvimento Científico e Tecnológico) grants 485472/2012-0 e 302525/2017-3 to C.A.F.S. and c (CAPES), Finance Code 001. S.R.C. had a doctoral fellowship from Fundação de Amparo à Pesquisa do Estado da Bahia, FAPESB. C.A.F.S., L.S.B and D.G had research productivity grants from CNPq."

"This work was supported by competitive grants "NEXTFRUT" grant # 0193.001.198/2016 from Fundação de Amparo à Pesquisa do Distrito Federal (FAP-DF) to DG, CNPq (Conselho Nacional de Desenvolvimento Científico e Tecnológico) grants 485472/2012-0 e 302525/2017-3 to C.A.F.S. and c (CAPES), Finance Code 001. S.R.C. had a doctoral fellowship from Fundação de Amparo à Pesquisa do Estado da Bahia, FAPESB. C.A.F.S., L.S.B and D.G had research productivity grants from CNPq."

6. Please review your reference list to ensure that it is complete and correct. If you have cited papers that have been retracted, please include the rationale for doing so in the manuscript text or remove these references and replace them with relevant current references. Any changes to the reference list should be mentioned in the rebuttal letter that accompanies your revised manuscript. If you need to cite a retracted article, indicate the article’s retracted status in the References list and also include a citation and full reference for the retraction notice.

Additional Editor Comments:

Reviewer #1:

The manuscript deals with the identification of genomic loci associated with resistance to Meloidogyne enterolobii in guava, the excellent fruit yielding tree species. It is an important research problem, where in several countries cultivate guava but the trees is routinely getting affected to nematodes. Marker assisted breeding is the suitable approach to produce pest resistant trees. The individuals were genotyped with EUChip60K containing Eucalyptus SNPs. Two SNP loci were found to be associated with nematode resistance. There are two publications available on the use of SNPs in Psidium and authors need to cite these references.

However, there is a complete and chromosome scale assembly of Psidium guajava with the coverage of about 95% has already been reported. The authors may utilise this genomic information and analyse SNP data generated to get the genomic coordinates of specific SNP regions identified from guava genome. Interpret the results based on the genomic regions of guava along with eucalypts.

The article is well written and reader friendly.

There are two publications available on the use of SNPs in guava, which need to be cited

1. Grossi, L.L., Fernandes, M., Silva, M.A. et al. DArTseq-derived SNPs for the genus Psidium reveal the high diversity of native species. Tree Genetics & Genomes 17, 23 (2021). https://doi.org/10.1007/s11295-021-01505-y

2. Diaz‐Garcia, Luis, and José S. Padilla‐Ramírez. "Development of single nucleotide polymorphism markers and genetic diversity in guava (Psidium guajava L.)." Plants, People, Planet (2022).

Reviewer #2:

In this manuscript an attempt has been made to identify the resistance gene for Meloidogyne enterolobii in Guava using EUChip60K SNPs genotyping and de novo genome assembly of the Psidium guajava. The SNP genotyping could transfer 14,268 SNP probe sets from Eucalyptus to Psidium at the nucleotide level. Two SNPs (EuBR03s29615246 and EuBR03s30383415) on chromosome 3 in a pseudo-assembly of Psidium guajava genome built using a syntenic path approach with the Eucalyptus grandis genome were identified linked to resistance locus for RKN. This contribution should advance the development of new guava cultivars through the development of inexpensive SNP based assays to monitor the introgression of the P. guineense-derived resistance for RKN into guava cultivars and rootstocks.

Reviewers' comments:

Reviewer's Responses to Questions

**Comments to the Author**

1. Is the manuscript technically sound, and do the data support the conclusions?

Reviewer #1: Yes

Reviewer #2: Yes

2. Has the statistical analysis been performed appropriately and rigorously? 

Reviewer #1: Yes

Reviewer #2: Yes

3. Have the authors made all data underlying the findings in their manuscript fully available?

Reviewer #1: Yes

Reviewer #2: Yes

4. Is the manuscript presented in an intelligible fashion and written in standard English?

Reviewer #1: Yes

Reviewer #2: Yes

5. Review Comments to the Author

Reviewer #1: The manuscript deals with the identification of genomic loci associated with resistance to Meloidogyne enterolobii in guava, the excellent fruit yielding tree species. It is an important research problem, where in several countries cultivate guava but the trees is routinely getting affected to nematodes. Marker assisted breeding is the suitable approach to produce pest resistant trees. The individuals were genotyped with EUChip60K containing Eucalyptus SNPs. Two SNP loci were found to be associated with nematode resistance. There are two publications available on the use of SNPs in Psidium and authors need to cite these references.

However, there is a complete and chromosome scale assembly of Psidium guajava with the coverage of about 95% has already been reported. The authors may utilise this genomic information and analyse SNP data generated to get the genomic coordinates of specific SNP regions identified from guava genome. Interpret the results based on the genomic regions of guava along with eucalypts.

The article is well written and reader friendly.

There are two publications available on the use of SNPs in guava, which need to be cited

Reviewer #2: In this manuscript an attempt has been made to identify the resistance gene for Meloidogyne enterolobii in Guava using EUChip60K SNPs genotyping and de novo genome assembly of the Psidium guajava. The SNP genotyping could transfer 14,268 SNP probe sets from Eucalyptus to Psidium at the nucleotide level. Two SNPs (EuBR03s29615246 and EuBR03s30383415) on chromosome 3 in a pseudo-assembly of Psidium guajava genome built using a syntenic path approach with the Eucalyptus grandis genome were identified linked to resistance locus for RKN. This contribution should advance the development of new guava cultivars through the development of inexpensive SNP based assays to monitor the introgression of the P. guineense-derived resistance for RKN into guava cultivars and rootstocks.

6. PLOS authors have the option to publish the peer review history of their article (what does this mean?). If published, this will include your full peer review and any attached files.

Reviewer #1: **Yes: **Ramasamy Yasodha

Reviewer #2: **Yes: **Dr Rakesh Singh

---

## [Author Response · Author response to Decision Letter 0]

11 Oct 2022

Below we respond to the reviewers’ comments.

Reviewer #1

REVIEWER COMMENT: The manuscript deals with the identification of genomic loci associated with resistance to Meloidogyne enterolobii in guava, the excellent fruit yielding tree species. It is an important research problem, where in several countries cultivate guava but the trees is routinely getting affected to nematodes. Marker assisted breeding is the suitable approach to produce pest resistant trees. The individuals were genotyped with EUChip60K containing Eucalyptus SNPs. Two SNP loci were found to be associated with nematode resistance. There are two publications available on the use of SNPs in Psidium and authors need to cite these references.

RESPONSE: Thank you for pointing out those two papers. We have now included the two references in the manuscript. In the interest of the readership, we added a comment regarding the fact that these two SNP genotyping experiments were carried out using genotyping by sequencing methods based on genome complexity reduction by digestion with restriction enzymes. Although such methods do offer advantages as far as no upfront cost and simultaneous SNPs discovery and genotyping, they are well known to have low reproducibility across independent experiments and laboratories, and suffer with large proportions of missing data. As such, although SNPs are genotyped, they do not constitute a legacy genomic resource for future widespread and continuous use by the community, nor they allow consolidation of data across studies in the same manner as the chip-based SNPs data we have used in our work.

REVIEWER COMMENT: However, there is a complete and chromosome scale assembly of Psidium guajava with the coverage of about 95% has already been reported. The authors may utilize this genomic information and analyze SNP data generated to get the genomic coordinates of specific SNP regions identified from guava genome. Interpret the results based on the genomic regions of guava along with eucalypts.

RESPONSE: We are fully aware of the chromosome scale assembly of the Psidium guajava genome. In fact, we have cited that paper already in the introduction when presenting the available genomic resources for the species. As the reviewer might have noticed, in our work we have generated a de novo assembly of the Psidium guajava genome for a different Psidium guajava tree than the one for which the genome had been published. We did exactly what the reviewer is suggesting, i.e., we used the genome assembly we generated to transfer a large number of SNP probesets that had been developed for Eucalyptus to the Psidium genome at the nucleotide level providing the exact genomic coordinates. Please also note that we have deposited our genome assembly at DDBJ/ENA/GenBank under the accession JAGHRR000000000. The version described in this article is JAGHRR010000000. These resources were deposited under the BioProject ID PRJNA713343. The entire presentation of results, their interpretation and discussion were therefore based on a detailed mapping of the SNP probesets on the Psidium genome we have generated, which is fully syntenic to the previously published Psidium genome. Therefore, all the necessary genomic address information for the SNPs is made available to the readership for future use of the publicly available EUCHIP60K to further genotype Psidium genetic resources.

Reviewer #2:

REVIEWER COMMENT: In this manuscript an attempt has been made to identify the resistance gene for Meloidogyne enterolobii in Guava usingEUChip60K SNPs genotyping and de novo genome assembly of the Psidium guajava. The SNP genotyping could transfer 14,268 SNP probe sets from Eucalyptus to Psidium at the nucleotide level. Two SNPs (EuBR03s29615246 andEuBR03s30383415) on chromosome 3 in a pseudo-assembly of Psidium guajava genome built using a syntenic path approach with the Eucalyptus grandis genome were identified linked to resistance locus for RKN. This contribution should advance the development of new guava cultivars through the development of inexpensive SNP based assays to monitor the introgression of the P. guineense-derived resistance for RKN into guava cultivars and rootstocks.

RESPONSE: We thank the reviewer for the positive comment.

---

## [Editor Report · Decision Letter 1]

17 Oct 2022

Genetic associations with resistance to Meloidogyne enterolobii in guava (Psidium sp.) using cross-genera SNPs and comparative genomics to Eucalyptus highlight evolutionary conservation across the Myrtaceae

PONE-D-22-23102R1

Dear Dr. Grattapaglia,

We’re pleased to inform you that your manuscript has been judged scientifically suitable for publication and will be formally accepted for publication once it meets all outstanding technical requirements.

Kind regards,

Zhenhai Han, PhD

Academic Editor

PLOS ONE
---

## [Editor Report · Acceptance letter]

20 Oct 2022

PONE-D-22-23102R1 

Genetic associations with resistance to *Meloidogyne enterolobii* in guava (*Psidium* sp.) using cross-genera SNPs and comparative genomics to *Eucalyptus* highlight evolutionary conservation across the Myrtaceae 

Dear Dr. Grattapaglia:

I'm pleased to inform you that your manuscript has been deemed suitable for publication in PLOS ONE. Congratulations! Your manuscript is now with our production department. 

Kind regards, 

on behalf of

Dr. Zhenhai Han 

Academic Editor

PLOS ONE